

# Integrated studies of a regional ozone pollution synthetically affected by subtropical high and typhoon system in the Yangtze River Delta region, China

**Lei Shu[1], Min Xie[1*], Tijian Wang[1,2*], Pulong Chen[1], Yong Han[1], Shu Li[1], Bingliang Zhuang[1],**

**Mengmeng Li[1], Da Gao[1]**

[1]School of Atmospheric Sciences, Nanjing University, Nanjing, China

[2]CMA-NJU Joint Laboratory for Climate Prediction Studies, Institute for Climate and Global

Change Research, School of Atmospheric Sciences, Nanjing University, Nanjing, China

---------------------------------------------------------------------------------------------------------

∗Corresponding author, +86-25-89685302

E-mail address: minxie@nju.edu.cn, tjwang@nju.edu.cn

**Abstract**: Severe high ozone ($O_3$) episodes usually have close relations to synoptic systems. A
regional continuous $O_3$ pollution episode is detected over the Yangtze River Delta (YRD) region in
China during August 7-12, 2013, in which the $O_3$ concentrations in more than half of the cities
exceeding the national air quality standard. The maximum hourly concentration of $O_3$ reaches
167.1 ppb. By means of the observational analysis and the WRF/CMAQ numerical simulation, the
characteristics and the essential impact factors of the typical regional $O_3$ pollution is integratedly
investigated. The observational analysis shows that the atmospheric subsidence dominated by
Western Pacific subtropical high plays a crucial role in the formation of high-level $O_3$. The
favorable weather conditions, such as extremely high temperature, low relative humidity and weak
wind speed, caused by the abnormal strong subtropical high are responsible for the trapping and
the chemical production of $O_3$ in the boundary layer. In addition, when the YRD cities at the front
of Typhoon Utor, the periphery circulation of typhoon system can enhance the downward airflows
and cause worse air pollution. But when the typhoon system weakens the subtropical high, the
prevailing southeasterly surface wind leads to the mitigation of the $O_3$ pollution. The Integrated
Process Rate (IPR) analysis incorporated in CMAQ is applied to further illustrate the combined
influence of subtropical high and typhoon system in this $O_3$ episode. The results show that the
vertical diffusion (VDIF) and the gas-phase chemistry (CHEM) are two major contributors to $O_3$





formation. During the episode, the contributions of VDIF and CHEM to $O_3$ maintain the high
values over 10 ppb/h in Shanghai, Hangzhou, and Nanjing. On August 10-11, the cities close to
the sea are apparently affected by the typhoon system, with the contribution of VDIF increasing to
28.45 ppb/h in Shanghai and 19.76 ppb/h in Hangzhou. When the YRD region is under the control
of the typhoon system, the contribution values of all individual processes decrease to a low level
in all cities. These results provide an insight for the $O_3$ pollution synthetically impacted by the
Western Pacific subtropical high and the tropical cyclone system.
**Keyword**: Ozone; subtropical high; typhoon; the Yangtze River Delta region; heat wave

**1. Introduction**
Ground-level ozone ($O_3$) is a secondary air pollutant generated by a series of complicated
photochemical reactions involving nitrogen oxides ($NO_x$) and hydrocarbons (HC) (Crutzen, 1973;
Sillman, 1999; Jenkin et al., 2000; Wang et al., 2006b; Xie et al., 2014; 2016b). Severe $O_3$
pollution events usually occur in the presence of sunlight and under favorable meteorological
conditions, with the abundance of $O_3$ precursors ($NO_x$ and HC) (Wang et al., 2006b). These $O_3$
pollutions in troposphere can deteriorate the air quality, and thereby cause adverse effects on
human health and vegetation (Feng et al., 2003; Fann and Risley, 2013; Landry et al., 2013).
Consequently, the formation mechanism and the integrated prevention of $O_3$ pollution are of great
concern in many megacities all over the world (Xie et al., 2016b).
Over the past decades, along with the rapid industrial and economic development, many areas
in China have been suffering from high levels of $O_3$ pollution. Especially in the most economically
vibrant and densely populated areas, such as the Yangtze River Delta (YRD) region, the Pearl
River Delta (PRD) region, and the Beijing-Tianjin-Hebei (BTH) area, the severe $O_3$ pollution
episode has frequently occurred (Lam et al., 2005; Wang et al., 2006b; An et al., 2007; Chan and
Yao, 2008; Duan et al., 2008; Jiang et al., 2008; Zhang et al., 2008; Guo et al., 2009; Shao et al.,
2009; Ma et al., 2012) , and the background air pollutant concentrations have steadily increased
(Chan and Yao, 2008; Zhang et al., 2008; Tang et al., 2009; Wang et al., 2009a; Ma et al., 2012;
Liu et al., 2013). Many studies on the $O_3$ pollution, including satellite data analyses, field
experiments, and model simulations, have been carried out over China in order to investigate the
temporal and spatial characteristics of surface photochemical pollutions (Lu and Wang, 2006;





Wang et al., 2006a; Tu et al., 2007; Zhang et al., 2007; 2008; Geng et al., 2008; Tang et al., 2008;
2009; Chen et al., 2009; Han et al., 2011; Ding et al., 2013; Xie et al., 2016b), nonlinear
photochemistry of $O_3$ and its precursors (Lam et al., 2005; Ran et al., 2009; Liu et al., 2010; Li et
al., 2011; Xie et al., 2014), interactions between $O_3$ and aerosols (Lou et al., 2014; Shi et al., 2015),
the effects of urbanization on $O_3$ formation (Wang et al., 2007; 2009b; Liao et al., 2015; Li et al.,
2016; Xie et al., 2016a; Zhu et al., 2016), and other essential impact factors (Jiang et al., 2012; Li
et al., 2012; Wei et al., 2012; Liu et al., 2013; Gao et al., 2016).
The Yangtze River Delta (YRD) region is a highly developed area of urbanization and
industrialization. With the accelerated economic development and remarkable increase in energy
consumption, the photochemical smog with high level of $O_3$ concentration is becoming more and
more prominent and frequent, tending to present conspicuous regional characteristics (Chan and
Yao, 2008; Ma et al., 2012; Li et al., 2012). Being located on the southeastern coast of China,
YRD features a typical subtropical monsoon climate and is strongly affected by the Western
Pacific subtropical high in summer. So, high $O_3$ concentrations are usually observed in late spring
and summer by in-situ monitoring (Ding et al., 2013; Xie et al., 2016b). Severe high $O_3$ episodes
usually have close relations to synoptic systems (Huang et al., 2005; 2006; Wang et al, 2006b;
Jiang et al., 2008; Cheng et al., 2014; Hung and Lo, 2015). Horizontal and vertical transport
processes from upwind $O_3$-rich air masses as well as poor atmospheric diffusion conditions can
lead to the accumulation of surface $O_3$ concentrations and aggravating the photochemical pollution
(Wang et al., 2006b). In previous studies on high $O_3$ pollution in the YRD region, some
researchers have discussed this issue. For example, Jiang et al. (2012) investigated the spring $O_3$
formation over East China, and suggested that $O_3$ concentrations over the YRD region were
transported and diffused from surrounding areas. Li et al. (2014) presented quantitative analysis on
atmospheric processes affecting $O_3$ concentrations in the typical YRD cities during a summertime
regional high $O_3$ episode, and found that the maximum concentration of photochemical pollutants
was usually related with the process of transportation. Gao et al. (2016) evaluated the $O_3$
concentration during a frequent shifting wind period, and revealed that vertical mixing played an
important positive role in the formation of surface $O_3$. However, these investigations only focused
on the $O_3$ formation mechanism for one megacity (such as Shanghai, Nanjing and Hangzhou, etc.)
or just a single station. Up to now, studies on the process analysis of high ozone episodes over the



YRD are quite limited (Li et al., 2012). So, more studies should pay attention to the typical
weather systems and the exact formation mechanism of the regional $O_3$ pollution in this region.
During August 7-12 2013, there is a typical regional $O_3$ pollution episode in the YRD region,
which may be combinedly influenced by the Western Pacific subtropical high and Typhoon Utor.
To fill the knowledge gap and better understand the important factors impacting $O_3$ formation
from the regional scale, we perform an observational analysis to identify the temporal and spatial
characteristics of the episode. With the aid of the WRF/CMAQ as well as the Integrated Process
Rate analysis (IPR) coupled within CMAQ, numerical simulations are conducted to provide
qualitative and quantitative analysis on the contributions of individual atmospheric processes. The
results may be a great help for the prediction and the prevention of high $O_3$ pollution events. In
this paper, the brief description of observational data and model configurations are shown in
Section 2. The detailed observational analysis of air quality and meteorological conditions are
given in Section 3. The evaluation of model performance and the formation mechanism of $O_3$
explored by IPR technique are presented in Section 4. In the end, a summary of main findings is
given in Section 5.

**2. Methodology**
**2.1 Observed meteorological and chemical data**
The weather charts and the observed surface meteorological records are used to analyze the
synoptic systems during the episode in August 2013, as well as to evaluate the model results of
meteorological factors. The weather charts for East Asia are accessible from Korea Meteorological
Administration. The hourly meteorological data at the observation sites of SH (31.40°N,121.46°E)
located in Shanghai, HZ (30.23°N, 120.16°E) in Hangzhou, and NJ (32.00°N, 118.80°E) in
Nanjing can be obtained from the University of Wyoming, where 2-m air temperature, 2-m
relative humidity, 10-m wind speed and10-m wind direction are available.
The air quality observational data are used to identify the regional characteristics of the $O_3$
episode and to validate the model performance for air pollutants. Fifteen cities are selected as the
representative research objects to better reflect the status of $O_3$ pollution over the YRD region. The
locations of these cities are shown in Fig. 1b, which contains Shanghai, 8 cities in Jiangsu
province (Changzhou, Nanjing, Nantong, Suzhou, Taizhou, Wuxi, Yangzhou, and Zhenjiang), and




6 cities in Zhejiang province (Hangzhou, Huzhou, Jiaxing, Ningbo, Shaoxing, and Zhoushan). The
in-situ monitoring data for the hourly concentrations of $O_3$, CO, $NO_2$, $SO_2$, $PM_{2.5}$ and $PM_{10}$ can be
acquired from National Environmental Monitoring Center (NEMC). The assurance/quality control
(QA/QC) procedures for monitoring strictly follow the national standards (State Environmental
Protection Administration of China, 2006). The hourly pollutant concentration for a city is
calculated as the average of the pollutant concentrations from several national monitoring sites in
that city, which can better characterize the pollution level of the city. In order to identify invalid or
lacking data, a checking procedure for these data is performed following the work of Chiqueto and
Silva (2010). Finally, only less than 0.2% of the primary data are ignored in the calculation.
**2.2 Model description and configurations**

WRF/CMAQ, which consists of the Weather Research and Forecasting (WRF) model version

3.4.1 and the Community Multi-scale Air Quality (CMAQ) Model version 4.7.1, is applied to
simulate the high $O_3$ episode over the YRD region in August 2013. WRF is a new generation of
meso-scale weather forecast model and assimilation system developed at the National Center for
Atmospheric Research (NCAR). Numerous applications have proven that it shows a good
performance in all kinds of weather forecasts and has broad application prospects in China (Jiang
et al., 2008; 2012; Wang et al., 2009b; Liu et al., 2013; Xie et al., 2014; 2016a; Liao et al., 2014;
2015; Li et al., 2016; Zhu et al., 2016). WRF provides off-line meteorological fields as the input
for the chemical transport model CMAQ. The CMAQ modeling system is a third generation of
regional air quality model developed by the Environmental Protection Agency of USA (USEPA).
A set of up-to-date compatible modules and control equations for the atmosphere is incorporated
in the model, which can fully consider atmospheric complicated physical processes, chemical
processes and the relative contribution of different species (Byun and Schere, 2006; Foley et al.,
2010). CMAQ has been widely applied in China and proven to be a reliable tool in simulating air
quality from city scale to meso scale (Li et al., 2012; Wei et al., 2012; Liu et al., 2013; Zhu et al.,

2016).

The simulation run is conducted from 08:00 (LST) on August 2nd to 08:00 (LST) on August

16th 2013, in which the first 48 h is taken as the spin-up time. Three one-way nested domains are
used in WRF with a Lambert Conformal map projection. The domain setting is shown in Fig.1.
The outermost domain (domain 1, d01) covers the most areas of East Asia and South Asia, with





the horizontal grids of 88×75 and the grid spacing of 81km. The nested domain d02 covers the
southeastern part of China, with the horizontal grids of 85×70 and the grid spacing of 27km. The
finest domain (domain 3, d03) covers the core areas of the YRD region, with the grid system of
70×64 and the resolution of 9km. For all domains, there are 23 vertical sigma layers from the
surface to the top pressure of 100hPa, with about 10 layers in the PBL. The detailed configuration
options for the dynamic parameterization in WRF are summarized in Table 1. Additionally, the
SLAB scheme that does not consider urban canopy parameters is adopted to model the urban
effect. In order to reflect the rapid urban expansion in the YRD region, the default USGS land-use
archives are updated by adding the present urban land-use conditions from 500-m Moderate
Resolution Imaging Spectroradiometer (MODIS) data, based on the work of Liao et al. (2014;
2015). The initial meteorological fields and boundary conditions are from NCEP FNL global
reanalysis data with $1^{o} \times 1^{o}$ resolution. The boundary conditions are forced every 6 h.

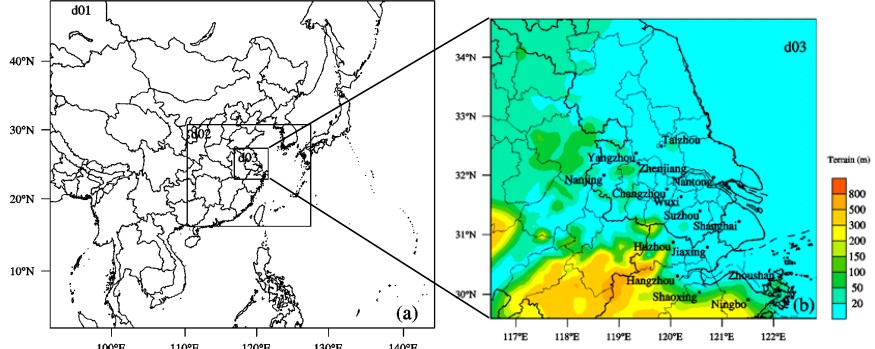


**Fig. 1. Domain settings, include (a) the three nested modeling domains and (b) the nested domain 3 (d03)**
**with the terrain elevations and the locations of 15 main cities in the YRD region.**

**Table 1. The grid settings and the physical options for WRF in this study.**

| Items | Options |
| --- | --- |
| Dimensions(x, y) | (88, 75), (85, 70), (70, 64) |
| Grid spacing (km) | 81, 27 ,9 |
| Microphysics | WRF Single-Moment 5-class scheme (Hong et al., 2004) |
| Longwave Radiation | RRTM scheme (Mlawer et al., 1997) |




| Shortwave Radiation | Goddard scheme (Kim and Wang, 2011) |
| Surface layer | Moni-Obukhov scheme (Monin and Obukhov, 1954) |
| Land-surface layer | Noah Land Surface Model (Chen and Dudhia, 2001) |
| Planetary Boundary layer | YSU scheme (Hong et al., 2006) |
| Cumulus Parameterization | Grell-Devenyi ensemble scheme (Grell and Devenyi, 2002) |


With respect to the air quality model, CMAQ uses the same vertical levels and the similar
three nested domains as those adopted in the meteorological simulation, whereas the CMAQ
domains are one grid smaller than the WRF domains. The Meteorology Chemistry Interface
Processor (MCIP) is used to convert WRF outputs to the input meteorological files needed by
CMAQ. The Carbon Bond 05 chemical mechanism (CB05) (Yarwood et al., 2005) is chosen for
gas-phase chemistry, and the 4rd generation CMAQ aerosol module (Byun and Schere, 2006) is
adopted for aerosol chemistry. The initial and outmost boundary conditions are obtained from the
Model for Ozone and Related Chemical Tracers version 4 (MOZART-4) (Emmons et al., 2010),
while those for the two nested inner domains are extracted from the immediate concentration files
of their parent domains. The anthropogenic emissions are mainly from the 2012-year
Multi-resolution Emission Inventory for China (MEIC) with 0.25°× 0.25° resolution, which is
re-projected for the grids of China in both domains. For the grids outside of China, the inventory
developed for the Intercontinental Chemical Transport Experiment-Phase B (INTEX-B) by Zhang
et al. (2009) is used. The natural $O_3$ precursor emissions are calculated by the natural emission
model developed by Xie et al. (2007; 2009; 2014), including NO from soil, VOCs from
vegetations, and $CH_4$ from rice paddies and terrestrial plants. The biomass burning emissions are
acquired from the work of Xie et al. (2014; 2016a).
**2.3 Integrated Process Rate (IPR) analysis method**
The CMAQ modeling system contains process analysis module (PROCAN), which consists
of the Integrated Process Rate (IPR) analysis and the Integrated Reaction Rate (IRR) analysis
(Byun and Schere, 2006). IPR has the capability of calculating the hourly contributions of
individual physical processes and the net effect of chemical reaction compared to the overall
concentrations, and thereby can determine the quantitative contribution of each process in a
specific grid cell. The atmospheric processes taken into consideration in IPR include the
horizontal advection (HADV), the vertical advection (ZADV), the horizontal diffusion (HDIF),





the vertical diffusion (VDIF), the emissions (EMIS), the dry deposition (DDEP), the cloud
processes with the aqueous chemistry (CLDS), the aerosol processes (AERO) and the gas-phase
chemistry (CHEM). The IPR analysis has been widely applied to investigate the regional
photochemical pollutions, and proven to be an effective tool to show the relative importance of
every process and provide a fundamental interpretation (Goncalves et al., 2009; Li et al., 2012; Liu
et al., 2013; Zhu et al., 2016).

In this paper, the period during August 4-15 is selected for the IPR analysis. With the aid of

IPR, we assess the roles of the individual physical and chemical processes involved in $O_3$
formation over the YRD region, and further present those in the typical cities such as Shanghai,
Nanjing and Hangzhou. Shanghai is the most populous city in China and Asia, as well as a global
financial and transportation center. Locating to the northwest of Shanghai, Nanjing is the capital of
Jiangsu Province and the second largest commercial center in East China. Hangzhou is the capital
of Zhejiang Province and located to the southwest of Shanghai. These cities are all highly
urbanized and industrialized, and suffer from severe $O_3$ pollution.
**2.4 Evaluation method**

Meteorological and air quality observation data are used to validate the reliability of

simulation in this study. Comparisons of the modeling results in the finest domain (d03) with the
hourly observation data are performed in Shanghai (31.40°N,121.46°E), Hangzhou (30.23°N,
120.16°E) and Nanjing(32.00°N, 118.80°E) for 2-m air temperature, 2-m relative humidity,
surface $O_3$ and surface $NO_2$. Additionally, the modeling results and observations for the surface
hourly $O_3$ concentrations in Wuxi (31.62°N, 120.27°E) is compared as well. The correlation
coefficient (R), the normalized mean bias (NMB) and the root-mean-square error (RMSE) are
used to evaluate the model performance. These statistic values are calculated as follows:
$$R = \frac{\sum_{i=1}^{N}(S_i - \bar{S})(O_i - \bar{O})}{\sqrt{\sum_{i=1}^{N}(S_i - \bar{S})^2}\sqrt{\sum_{i=1}^{N}(O_i - \bar{O})}} \qquad (1)$$
$$NMB = \frac{\sum_{i=1}^{N}(S_i - O_i)}{\sum_{i=1}^{N}O_i} \times 100\% \qquad (2)$$
$$RMSE = [\frac{1}{N}\sum_{i=1}^{N}(S_i - O_i)^2]^{\frac{1}{2}} \qquad (3)$$





Where $S_i$ represents the simulated value and $O_i$ represents the observed value. $N$ means the total
number of valid data. Generally, the model performance is acceptable if the values of NMB and
RMSE are close to 0 and that of R is close to 1.

**3. Characteristics of the continuous ozone episode**
**3.1 Basic characteristic of the regional ozone episode in August 2013**

Fig. 2 shows the temporal variation of the hourly $O_3$ concentrations observed by in-situ

monitoring in 15 typical cities over the YRD region from 00:00 (UTC) 4 August to 23:00 (UTC)
15 August in 2013. Obviously, from August 7 to August 12, high $O_3$ concentrations over 93.5 ppb
(approximately equal to the national air quality standard of 200 $\mu g/m^3$ for the hourly $O_3$
concentration) have been frequently recorded in 13 cities, which means most cities over the YRD
region exceed the national air quality standard. So, this high $O_3$ pollution episode is a typical
regional $O_3$ pollution episode that can affect the people and the ecosystem in a large area. Table 2
presents the highest and the average concentrations of $O_3$, as well as its precursors ($NO_2$ and CO),
observed in these 15 cities during August 7- 12 2013. The highest hourly $O_3$ concentration occurs
in Nantong with the value of 167.1 ppb, which is nearly 2 times of the national air quality standard,
followed by 166.1 and 162.4 ppb in Changzhou and Jiaxing, respectively. It seems that $O_3$
concentrations are higher in the cities around Shanghai, where the concentrations of $O_3$ precursors
(shown in Table 2) and the water vapor are more adequate as well. High concentrations of $O_3$ and
its precursors imply that there may be stronger photochemical reactions in these cities.



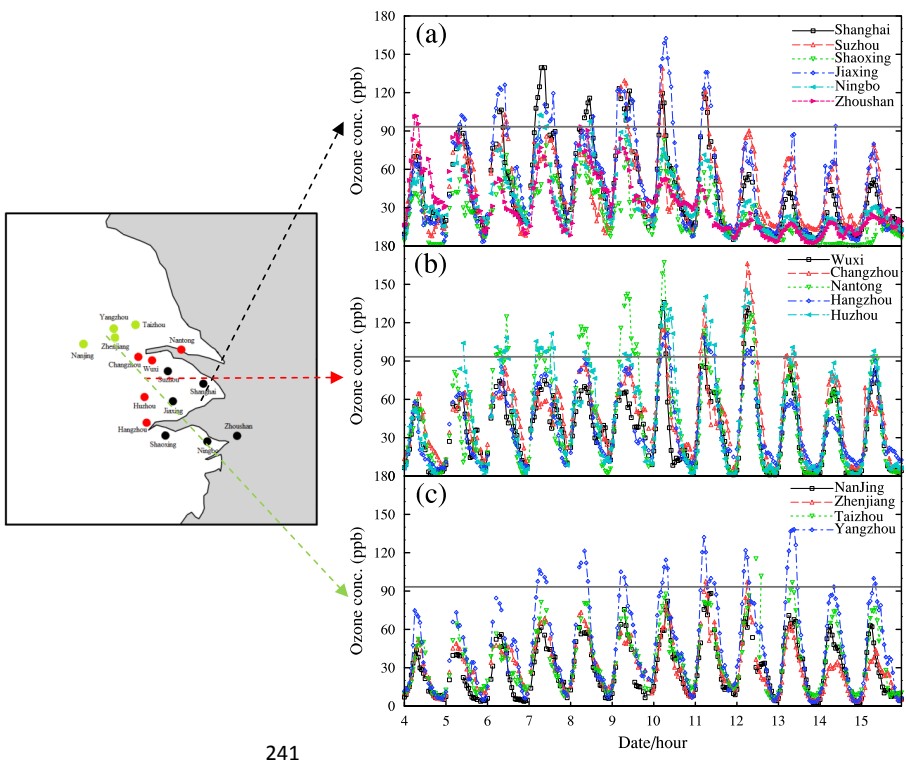


**Fig. 2. The time series of the observed O$_3$ concentrations in 15 typical cities from4 to 15 August 2013 over the YRD region, which can be divided into three areas: (a) the Southeast Coast Region (SCR), including Shanghai, Suzhou, Shaoxing, Jiaxing, Ningbo, and Zhoushan; (b) the Central Inland Region (CIR), including Wuxi, Changzhou, Nantong, Hangzhou, and Huzhou; (c) the Northwest Inland Region (NIR), including Nanjing, Zhenjiang, Taizhou and Yangzhou. The gray solid lines in (a), (b), and (c) represent the national standard for the hourly O$_3$ concentration, which is 200 μg/m$^3$.**

**Table 2. The maximum and average concentrations of O$_3$, NO$_2$, and CO observed in 15 cities (ppb)**

| Sites | | O$_3$ | | NO$_2$ | | CO | |
|---|---|---|---|---|---|---|---|
| | | Max | Mean | Max | Mean | Max | Mean |
| Southeast Coast Region (CSR) | Shanghai | 139.5 | 55.1 | 35.1 | 15.6 | 1184.0 | 605.5 |
| | Suzhou | 139.1 | 50.9 | 50.6 | 19.7 | 904.0 | 567.2 |
| | Jiaxing | 162.4 | 61.1 | 52.1 | 17.1 | 1128.0 | 671.6 |
| | Ningbo | 113.4 | 41.9 | 31.2 | 12.4 | 784.0 | 566.3 |
| | Shaoxing | 82.6 | 31.9 | 27.8 | 12.7 | 880.0 | 635.9 |
| | Zhoushan | 93.6 | 35.5 | 27.3 | 7.8 | 680.0 | 460.6 |
| Central Inland Region | Hangzhou | 111.5 | 48.6 | 30.2 | 16.7 | 712.0 | 472.1 |
| | Huzhou | 145.6 | 57.2 | 43.8 | 20.8 | 1040.0 | 661.3 |
| | Wuxi | 135.8 | 43.2 | 39.9 | 18.8 | 1824.0 | 785.3 |





| | | | | | | |
|---|---|---|---|---|---|---|
| (CIR) | Changzhou | 166.1 | 55.7 | 58.4 | 24.5 | 1880.0 | 719.0 |
| | Nantong | 167.1 | 56.0 | 48.2 | 20.9 | 1224.0 | 655.4 |
| Northwest | Nanjing | 88.2 | 34.1 | 41.4 | 21.9 | 1640.0 | 813.9 |
| Inland | Yangzhou | 132.1 | 54.1 | 36.0 | 17.1 | 1568.0 | 710.1 |
| Region | Zhenjiang | 97.5 | 37.7 | 38.5 | 20.1 | 1752.0 | 963.0 |
| (NIR) | Taizhou | 115.3 | 40.5 | 18.5 | 7.7 | 1640.0 | 1094.0 |

According to the temporal variation characteristics of $O_3$ illustrated in Fig. 2, the abovementioned 15 typical YRD cities can be classified into three categories: (1) the cities in the Southeast Coastal Region (SCR), including Shanghai, Suzhou, Jiaxing, Ningbo, Shaoxing, and Zhoushan; (2) the cities in the Central Inland Region (CIR), including Hangzhou, Huzhou, Wuxi, Changzhou, and Nantong; and (3) the cities in the Northwestern Inland Region (NIR), including Nanjing, Yangzhou, Zhenjiang, and Taizhou. The classification is primarily on basis of the observational facts that the maximum $O_3$ concentrations occur on August 10-11, 12, and 13, and begin to synchronously decrease on August 12, 13 and 14 in SCR, CIR and NIR, respectively. As shown in Fig. 2, in the Southeast Coastal Region (SCR), Zhoushan firstly exceeds the national $O_3$ standard on August 4th, followed by Jiaxing, Shanghai, Suzhou and Ningbo. The peak hourly $O_3$ concentration of SCR occurs in Jiaxing on August 10 with the value up to 162.4 ppb. In the Central Inland Region (CIR), Huzhou is the first city exceeding the national $O_3$ standard, followed by the order of Nantong, Changzhou, Wuxi and Hangzhou. The high-level $O_3$ pollution in Huzhou lasts from August 5th to 13th. In Nantong and Changzhou, the maximum hourly $O_3$ concentration reaches 167.1 ppb on August 10 and 166.1 ppb on August 12, respectively. As for the Northwest Inland Region (NIR), Yangzhou, Zhenjiang and Taizhou successively exceed the national $O_3$ standard. It is also noteworthy that the date when $O_3$ concentration exceed the national air quality standard in coastal region is ahead of that in inland regions, so is the date of $O_3$ decrease. The different time of the $O_3$ decrease in different regions might be related to the strong southeast wind in accordance with the movement of Typhoon Utor, which is discussed in Sect. 3.2 in detail.

In general, for each city, there is a remarkable continuous growth in $O_3$ concentrations before the $O_3$ episode, followed by the lasting heavy $O_3$ pollution period. Though the $O_3$ concentrations in Shaoxing and Nanjing meet the national $O_3$ standard, their time series still show the similar tendency for the other cities in the same region. The excessive level of $O_3$ occurring in Huzhou,


Jiaxing, Nantong, Yangzhou and Shanghai lasts for more than six consecutive days, reflecting the
regional continuous characteristics of this $O_3$ pollution episode. As for $NO_2$ and CO, their average
concentrations in the YRD region during the $O_3$ episode show the variation range of
approximately 7.7~24.5 and 460.0~1094.0 ppb, respectively, indicating the heterogeneity of the
spatial emission distribution of $O_3$ precursors. Besides, the relative high hourly concentrations of
$NO_2$ show good agreements with those of $O_3$, implying it is one of the important $O_3$ precursor.

**3.2 Meteorological condition and its effect**

Favorable weather conditions have large impacts on the formation of severe $O_3$ pollution
(Huang et al., 2005; 2006; Wang et al, 2006b; Jiang et al., 2008; Cheng et al., 2014; Hung and Lo,
2015). High-level $O_3$ episodes often take place in hot seasons, when the meteorological conditions
with high temperature and strong solar radiation are beneficial to the photochemical reactions of
$O_3$ (Lam et al., 2005). Fig. 3 shows the variations of the surface meteorological parameters that are
related to this photochemical pollution episode during August 4-15, including 2-m air temperature,
2-m relative humidity, 10-m wind speed and 10-m wind direction at the meteorological sites of SH
(31.40°N,121.46°E) located in Shanghai of SCR, HZ (30.23°N, 120.16°E) located in Hangzhou of
CIR, and NJ (32.00°N, 118.80°E) located in Nanjing of NIR .
As shown in Fig. 3a, the hot weather at SH, HZ and NJ exists for nearly a week from August
7 to 12, with the hourly maximum temperature reaching the value over 40 ℃. Meanwhile, the
variations of 2-m relative humidity show the negative correlation with those of 2-m air
temperature. The minimum 2-m relative humidity at SH and HZ occur on August 9 and August 10
respectively, with the value below 75%. These minimum values are also lower than the values
before and after the $O_3$ episode, suggesting that high-level $O_3$ episodes usually occur under the
weather conditions with high temperature and low humidity. The value of 2-m relative humidity at
NJ is relatively higher than those at SH and HZ and remains more stable. This extremely hot and
dry weather condition at SH, HZ, and NJ are successively relieved on August 12, 13 and 15, which
coincide well with the reduction of surface $O_3$ concentrations in Shanghai, Hangzhou, and Nanjing
(Fig. 2). With respect to the observed surface wind (Fig. 3b), the 10-m wind speed at SH, HZ, and
NJ is comparatively lower during the period of the $O_3$ episode, while it is suddenly intensified
after August 12. Meanwhile, the wind direction is fluctuating from 7 to 12 August, while it
maintains southeasterly wind after August 12 as well. The growth of wind speed is more distinct at



SH, with the maximum value of approximately 9 m/s. The wind speed at NJ has an obviously
diurnal variation from August 4 to 8, and the minimum value occurs on August 10.

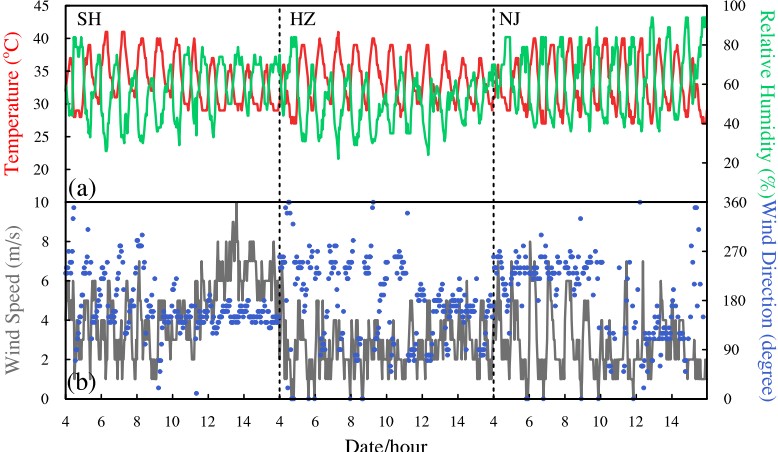


**Fig. 3. Temporal variations of the main meteorological parameters at SH (31.40°N,121.46°E), HZ (30.23°N,**

**120.16°E) and NJ (32.00°N, 118.80°E) meteorological stations during August 4-15, 2013: (a) 2-m air**

**temperature (the red solid line) and 2-m relative humidity (the green solid line); (b) 10-m wind speed (the**

**gray solid line) and 10-m wind direction (the blue scatter points).**


Fig. 4 displays the weather charts for the 500hPa layer over the East Asia at 00:00 (UTC) on
6, 8, 10, and 12 August 2013, which can illustrate the main synoptic patterns causing the $O_3$
pollution. Obviously, during the period of the selected $O_3$ episode, the whole YRD region is under
the control of the strong Western Pacific subtropical high, which might be the direct and leading
cause of the abnormal high temperature shown in Fig. 3a. The intensity of the subtropical high is
usually characterized by the area index, defined as the total number of grid points that have
geopotential heights of 588 decameters or greater in the region of 110-180°E and northward of
10°N. As shown in Fig. 4, the 588-decameter area covers most of southeast China, and the high
pressure center (592-decameter area) is located in the southeastern coastal areas as well as the
surrounding sea areas, which means the subtropical high is very intensive. This high pressure
strengthens and remains over the YRD region for several days (from August 6 to 12), implying
that the air subsides to the ground. The downward air acts as a dome capping the atmosphere, and
helps to trap heat as well as air pollutants at the surface. Without the lift of air, there is little



convection and therefore little cumulus clouds or rains. The end result is a continual accumulating
of solar radiation and heat on the ground, which may greatly enhance the photochemical reactions
between the abundant build-up air pollutants.

The other weather system worthy of note is Typhoon Utor (shown in Fig. 4c and d). Typhoon

Utor is one of the strongest typhoons in the 2013 Pacific typhoon season, with the international
code of 1311. It is formed early on August 8, develops into a tropical storm on August 9,
undergoes a explosive intensification within a half of day, and achieves typhoon status on early
August 10. After landing in Luzon of the Philippines on late August 11, it reemerges in the South
China Sea on August 12. Typhoon Utor hits the land of Guangdong Province in China on August
14, and thereby is finally weakened into a tropical storm. In the end, it is ultimately dissipated on
August 18. It was reported that ozone episodes during the hot season are usually associated with
the passage of tropical cyclones close to the territory (Huang et al., 2005; Wang et al., 2006b;
Jiang et al., 2008; Cheng et al., 2014; Hung and Lo, 2015). When a site is at the front of moving
typhoon system, it can be controlled by the downward airflow induced by the typhoons' peripheral
circulation. So, the typhoon system can cause the local weather around the site with high
temperature, low humidity, strong solar radiation and small wind for a short time, before it is close
enough to bring winds and rains. All these changes of meteorological conditions can significantly
affect the formation of severe continuous $O_3$ pollution (Jiang et al., 2008). In this $O_3$ episode, the
YRD region may be influenced by the peripheral circulation of Typhoon Utor as well. Especially
on August 10-11, the downward airflow in the troposphere is significantly strengthened (shown in
Fig. 6 and detailedly discussed in Sect. 4.2), which may enhance the build-up of heat and air
pollutants and thereby result in worse air pollution shown in Fig. 2.

Moreover, from August 12 to 14 (shown in Fig. 4d), with the approaching of Typhoon Utor,

the near-surface breeze over the YRD region gradually turns to be the prevailing southeasterly or
southerly wind, with the highest wind speed up to 6-10 m/s in Shanghai. The strengthened wind
can bring the clean marine air from ocean to inland, and thereby effectively mitigate the $O_3$
pollution. Meantime, Typhoon Utor also gradually affects the position and strength of the Western
Pacific subtropical high. As the typhoon continuous approaching and finally landing on
Guangdong, the high pressure system is forced to retreat easterly and move northwards. When the
high pressure center completely moves to the oceans, the YRD region is totally under the control





of the typhoon system. In the end, the hot weather is relieved and the $O_3$ pollution is mitigated.
The coastal cities in CSR are closer to the typhoon system, so they are firstly influenced during
this period. Thus, the wind at SH in CSR firstly changes, followed by HZ in CIR and NJ in NIR.
In the same way, 2-m air temperature and $O_3$ concentrations also successively decrease from
southeast (SH in CSR) to northwest (NJ in NIR) owing to the scavenging effect.

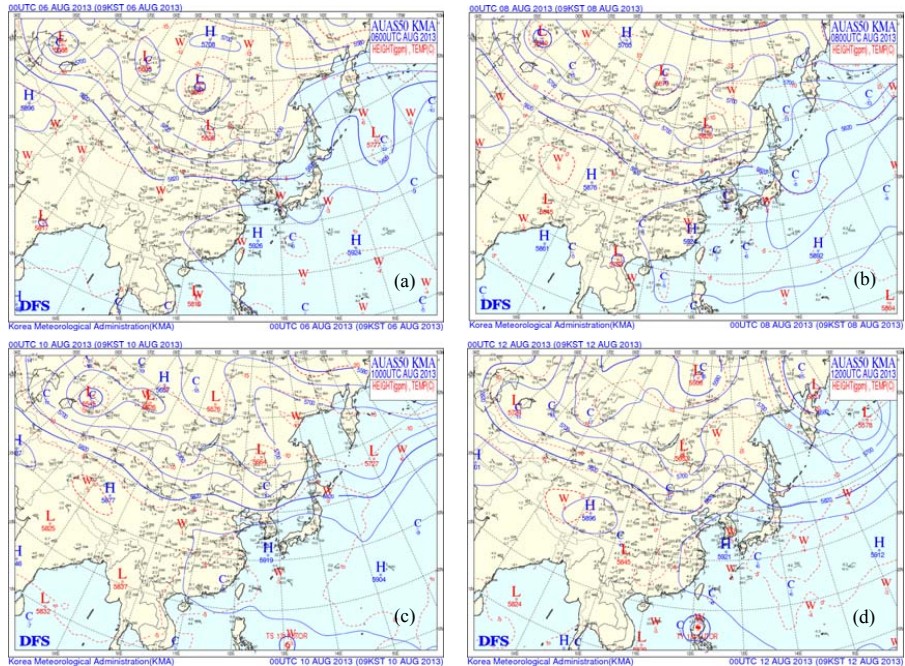

**Fig. 4. Weather charts at the 500hPa layer over the East Asia at 00:00 (UTC) on (a) August 6, (b) August 8,**
**(c) August 10, and (d) August 12 2013 (from Korea Meteorological Administration)**

**4 Modeling results and discussions**
**4.1 Evaluation of model performance**

To evaluate the simulation performance, the hourly modeling results during the period of

4-15 August 2013 are compared with the observation records. Table 3 presents the performance
statistics, including the values of the correlation coefficient (R), the normalized mean bias (NMB),
and the root-mean-square error (RMSE), which are all calculated for 2-m air temperature ($T_2$),
2-m relative humidity ($RH_2$), surface ozone concentrations ($O_3$), and surface nitrogen dioxide
concentrations ($NO_2$) in Shanghai (SH), Nanjing (NJ), and Hangzhou (HZ).





As indicated in Table 3, the simulated results of surface air temperature and relative humidity

from WRF show good correlation with the observations. The highest correlation coefficient of 2-m

air temperature ($T_2$) is found to be 0.91 at SH, followed by 0.84 at NJ and 0.80 at HZ (statistically

significant at 95% confident level). The corresponding correlation coefficients for 2-m relative

humidity ($RH_2$) are 0.85, 0.83 and 0.78, respectively. The values of RMSE for $T_2$ at SH, NJ and

HZ are 4.15, 2.91and 3.09$^o$C, and those for $RH_2$ are 19.3%, 9.41% and 13.96% respectively.

However, our simulation underestimates $T_2$ and overestimates $RH_2$ to some certain extent, with the

values of NMB for $T_2$ at SH, NJ and HZ being -11.69%, -5.98% and -6.53%, and those for $RH_2$

being 12.64%, 4.52% and 16.36%. These biases might be attributed to the uncertainty caused by

the SLAB scheme, which can underestimate temperature in summer (Liao et al., 2014). According

to the relevant studies (Li et al., 2012; Liao et al., 2015; Xie et al., 2016a), this level of over- or

under-estimation is still acceptable. In summary, the abovementioned performance statistics

numbers basically illustrate that the WRF simulation can reflect the major characteristics of

meteorological conditions during this $O_3$ episode, and the meteorological outputs can be used in

the pollutant concentration simulation.


**Table 3. Comparisons between the simulations and the observations at Shanghai, Nanjing and Hangzhou**

**stations.**

| Sites [a] | Vars [b] | Mean | | R [e] | NMB [f] | RMSE [g] |
|---|---|---|---|---|---|---|
| | | OBS [c] | SIM [d] | | | |
| SH | $T_2$ (°C) | 33.27 | 31.38 | 0.91 | -11.69% | 4.15 |
| | $RH_2$ (%) | 57.91 | 65.23 | 0.85 | 12.64% | 19.3 |
| | $O_3$ (ppb) | 87.77 | 82.5 | 0.81 | -6.00% | 38.79 |
| | $NO_2$ (ppb) | 29.01 | 38.25 | 0.54 | 38.75% | 28.95 |
| NJ | $T_2$ (°C) | 32.95 | 30.98 | 0.84 | -5.98% | 2.91 |
| | $RH_2$ (%) | 63.28 | 66.14 | 0.83 | 4.52% | 9.41 |
| | $O_3$ (ppb) | 69.7 | 78.15 | 0.81 | 26.47% | 36.8 |
| | $NO_2$ (ppb) | 41.44 | 40.09 | 0.61 | -3.26% | 22.4 |
| HZ | $T_2$ (°C) | 33.25 | 31.08 | 0.8 | -6.53% | 3.09 |
| | $RH_2$ (%) | 52.76 | 61.39 | 0.78 | 16.36% | 13.96 |
| | $O_3$ (ppb) | 76.57 | 84.51 | 0.83 | 10.37% | 33.95 |
| | $NO_2$ (ppb) | 31.06 | 27.21 | 0.66 | -12.40% | 16.86 |

[a] Sites indicates the city where the observation sites locate, including Shanghai (SH), Nanjing (NJ), and Hangzhou
(HZ); [b] Vars indicates the variables under validation, including 2-m air temperature ($T_2$), 2-m relative humidity
($RH_2$), ozone ($O_3$), and nitrogen dioxide ($NO_2$). The words between the parentheses behind variables indicate the



unit; $^c$ OBS indicates the observation data; $^d$ SIM indicates the simulation results from WRF/Chem; $^e$ R indicates
the correlation coefficients, with statistically significant at 95% confident level; $^f$ NMB indicates the normalized
mean bias; $^g$ RMSE indicates the root-mean-square error.

Fig. 5 shows the comparisons between the model results from CMAQ and the observed
hourly concentrations of $O_3$ in Shanghai, Nanjing, Hangzhou and Wuxi during 4-15 August 2013.
Obviously, the observations and the simulated results present reasonable agreement at each site,
with the correlation coefficients of 0.81 to 0.83, NMB of -6% to 26.47%, RMSE of 33.95 to 38.79
ppb. Moreover, the simulation also reproduces the diurnal variation of $O_3$, which shows that the
concentration reaches its maximum at around noon time and gradually decreases to its minimum
after midnight. With respect to the $O_3$ precursor, comparisons of $NO_2$ concentrations between
simulation results and observations show that the correlation coefficient at each city is about 0.6
(given in Table 3), which further prove that the process of $O_3$ formation is captured reasonable
well over the YRD region and throughout the episode. However, CMAQ overestimates $NO_2$ and
underestimates $O_3$ in Shanghai, while underestimates $NO_2$ and overestimates $O_3$ in Nanjing and
Hangzhou. These biases of $O_3$ and $NO_2$ can be attributed to the uncertainties related with $O_3$
precursor emissions, meteorology, and observation deviation (Li et al., 2012). Moreover, the
uncertainty in nonlinear chemical reactions coupled in CMAQ may also have important effects on
model predictions. For example, the modeling results cannot catch the low $O_3$ values observed at
night in Nanjing (Fig. 4b), Hangzhou (Fig. 4c) and Wuxi (Fig. 4d), implying there may be some
imperfections in the nocturnal chemistry of CMAQ. Nevertheless, the performance of CMAQ
model is comparable to the other applications (Goncalves et al., 2009; Li et al., 2012; Zhu et al.,
2016). Compared to these previous related studies, the simulation in this study attains an
acceptable and satisfactory result. Thus, the consistency of simulation and observation
demonstrates that the modeling results are capable of capturing and reproducing the characteristics
and changes of photochemical pollutants, and can be used to provide valuable insights into the
governing processes of this $O_3$ episode.



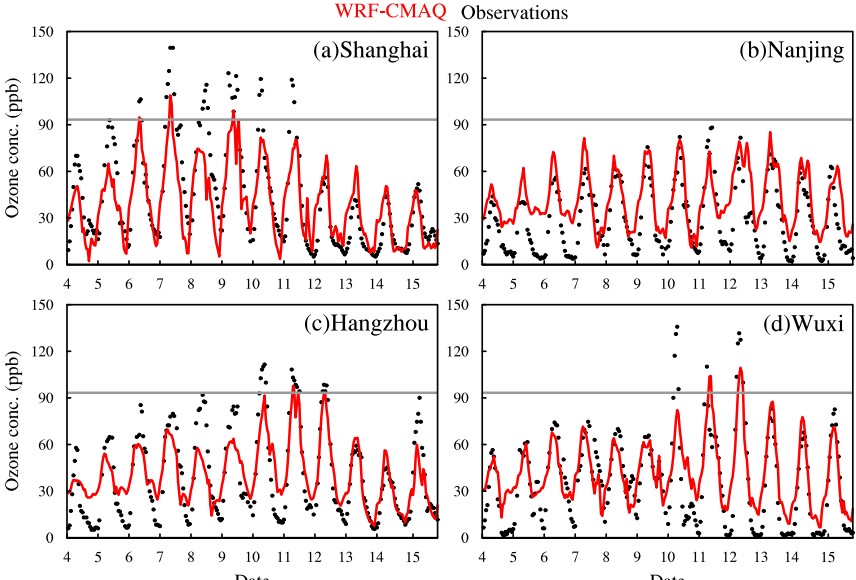


Fig. 5. Hourly variations of the observed and the simulated O$_3$ concentrations in (a) Shanghai (SH), (b)
Nanjing (NJ), (c) Hangzhou (HZ), and (d) Wuxi. In (a), (b), (c), and (d), the red solid lines show the
modeling results, the black dot lines give the observations, and the solid gray lines represent the national
standard for the hourly O$_3$ concentration, which is 200μg/m$^3$.

**4.2 Characteristics of the vertical airflows**

Fig. 6 presents the vertical wind velocity as well as the vertical distribution of O$_3$
concentrations from 29.6°E to 34.7°E during August 7 - 12 2013. Along the vertical cross-section,
the values from 118°E to 122°E are averaged in the meridional direction. The simulation results
clearly illustrate that there are strong downward airflows over the YRD region during the period of
the regional high-level O$_3$ pollution, which can be attributed to the fact that these areas are under
the control of the subtropical high and the sinking airflow is predominant (as discussed in Sect.
3.2).

From 7 to 9 August 2013 (shown in Fig. 6a-c), except for the mentioned regional sinking
airflows, there are still some local thermal circulations, which are related with urban heat islands,
continually occurring at the lower atmospheric layers (< 2km) along the vertical cross-section of
Hangzhou (HZ) - Shanghai (SH) - Nanjing (NJ). Usually high pressures are accompanied by more
stagnant and fair dry weather, so the upward and the downward flows caused by urban-breeze



circulations can easily appear in the urban areas of SH, HZ, and NJ. With respect to $O_3$, high
concentrations (> 60ppb) usually appear from the surface to 1.5km height, with the maximum
values over 70 ppb in and around cities. As discussed in Sect. 3.2, induced by the regional sinking
airflows, air pollutants are tend to be trapped on the ground. Moreover, the local circulations over
the cities (Fig. 6) make the urban areas to be the convergence zones, and thereby more air
pollutants can be accumulated in and around these cities. Under the weather conditions induced by
the subtropical high, such as high air temperature, stronger solar radiation and less water vapor,
the chemical reactions between the build-up air pollutants can be enhanced to form the high-level
$O_3$ pollution.
However, from 10 to 12 August, with the approaching of Typhoon Utor, the vertical air
movements over the YRD region are not restricted at the lower atmosphere any more. As shown in
Fig. 6d and e (August 10 and 11), there are stronger downward airflows from the surface to the top
of troposphere. As discussed in Sect. 3.2, SH, HZ, and NJ are at the front of the moving typhoon
system, so the peripheral circulation of Typhoon Utor may enhance the sinking of atmosphere,
which can lead to higher air temperature, lower humidity, and stronger solar radiation. Affected by
the enhanced downward air movement as well as the relevant changes of meteorological
conditions, $O_3$ concentrations over the YRD region maintain a high pollution level, with the $O_3$
concentrations over 60 ppb below the height of 1.5 km. Furthermore, as shown in Fig. 6d to f, the
high value center of $O_3$ concentrations moves from southeast to northwest during August 10 - 12,
implying that the peripheral circulation of Typhoon Utor can drive the air from the coastal areas to
the inland areas.





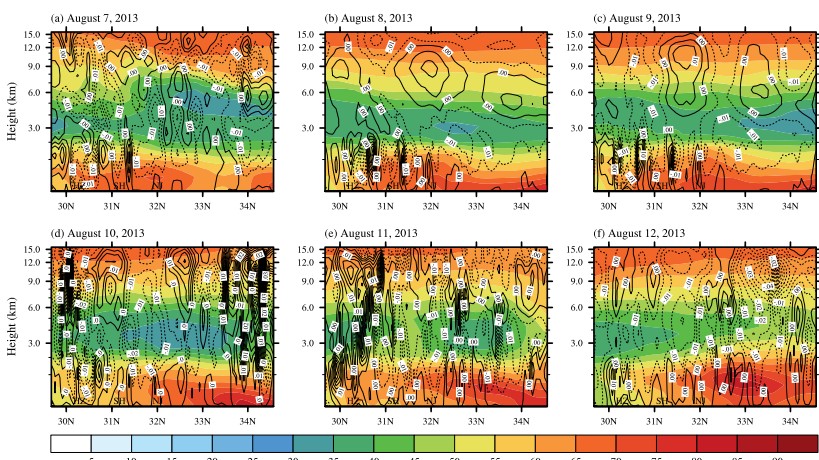


**Fig. 6. Simulated vertical wind velocity and vertical distribution of O₃ concentrations in the YRD region from 29.6°E to 34.7°E during 7 to 12 August 2013, with the values averaged in the meridional direction from 118°E to 122°E. The marks of HZ, SH, and NJ point out the latitudes of Hangzhou, Shanghai, and Nanjing, respectively.**


The vertical changes of wind velocity and O₃ concentrations above Shanghai are further
illustrated in Fig. 7. Similar to that in Fig. 6, the atmospheric subsidence can also be found in the
boundary layer of Shanghai during the period of the high-level O₃ pollution (from 7 to 12 August).
Affected by the extremely high temperature, more active photochemical reactions lead to higher
O₃ concentrations in the whole atmospheric boundary layer. The downward airflows induced by
the subtropical high trap and enhance the accumulation of surface O₃ as time passes. Thus, high
O₃ concentrations are formed below 2 km above the urban areas of Shanghai, and the high
concentration centers occur near the surface below 500 m. It is interesting that O₃ concentration on
August 8 is comparatively lower, which can be seen in Fig. 2 as well. This phenomenon can be
explained by the fact shown in Fig. 7 that the transient upward airflow occurs at above 300 m over
Shanghai and inhibits the accumulation of the O₃ pollution at the surface. Additionally, Fig. 7 also
presents the possible effects of Typhoon Utor on the formation of O₃. On August 10, when the
typhoon system approaches to the eastern coastal areas of China, the sinking air above Shanghai is
apparently strengthened, and thereby enhances the intensity of O₃ pollution as well as the scope of
the pollution. But after August 12, when Typhoon Utor changes the wind and even impacts the
subtropical high, high temperature is alleviated and the build-up O₃ is transported to other places.





Thus, the pollution is mitigated.

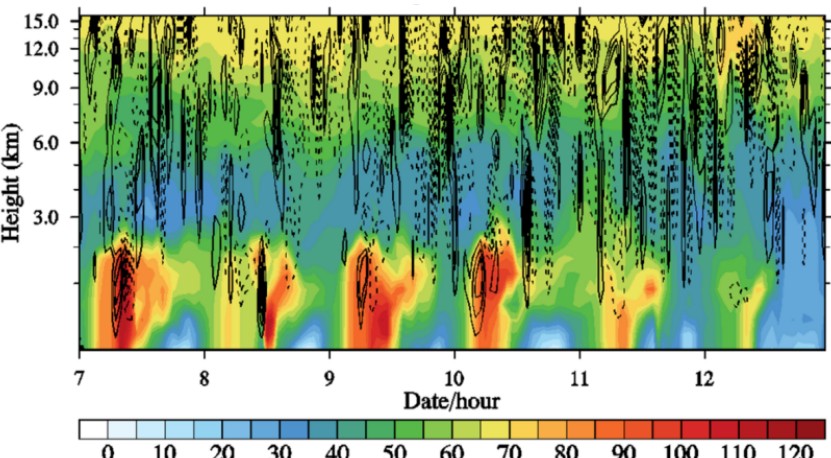


**Fig. 7. Temporal variations for the vertical wind velocity and the vertical distribution of O$_3$ concentrations**
**above Shanghai (SH) during August 7 to 12, 2013.**

**4.3 Process analysis for ozone formation**
**4.3.1 Typical cities in the YRD region**

Fig. 8 shows the daily contributions of different atmospheric processes to the formation of O$_3$

in Shanghai (SH), Nanjing (NJ), and Hangzhou (HZ) at the first modeling layer from 4 to 15
August 2013. As shown in the figure, for all cities during this period, the major contributors to
high O$_3$ concentrations include the vertical diffusion (VDIF), the dry deposition (DDEP), the
gas-phase chemistry (CHEM), and the total advection (TADV). TADV is the sum of the horizontal
advection (HADV) and the vertical advection (ZADV). In this study, HADV and ZADV are
considered together as TADV because they are inevitably linked as the inseparable parts of air
circulation. As discussed in Sect. 3.2, the strong sinking air causes slow wind on the ground and
little clouds in the sky, so the contributions of horizontal diffusion (HDIF) and cloud processes
(CLDS) are quite small during this episode.

In the first layer of the urban areas of Shanghai (Fig. 8a), the averaged daily contributions

during 4-15 August for the vertical diffusion (VDIF), the gas-phase chemistry (CHEM), the
advection processes (TADV) and the dry deposition (DDEP) are 9.95, 10.10, -11.74 and -7.28
ppb/h, respectively. Obviously, VDIF and CHEM exhibit significant positive contributions to O$_3$





during most days, while TADV and DDEP mainly show the consumption contributions. The
sinking air caused by the weather system discussed in Sect. 3.2 can trap heat and air pollutants on
the ground, and results in VDIF to be the most import source of surface $O_3$. Meanwhile, the hotter
and dryer weather with more sunshine, which is related with the sinking air, can enhance the
photochemical reactions. So, CHEM can form more $O_3$ on the ground. Compared with the time
series of CHEM and DDEP in which there are no obvious fluctuations, the values of VDIF and
TADV significantly change with the time, with the daily mean contributions varying from 3.99 to
28.45 ppb/h for VDIF and from -2.56 to -28.13 ppb/h for TADV. These time variations should be
related with the changes of vertical air movement. For example, the value of VDIF on August 8 is
only 3.99 ppb/h, which can be attributed to the local transient upward airflow over Shanghai
(shown in Fig. 7). On August 10, however, VDIF can contribute 28.45 ppb $O_3$ per hour, which
may be related with the enhanced downward air movement caused by the peripheral circulation of
Typhoon Utor. Moreover, during the high-level $O_3$ episode from August 7-12, the mean values for
VDIF, CHEM, TADV and DDEP are 13.41, 11.21, -8.37 and -14.74 ppb/h. But after August 12,
the mean contributions of VDIF, CHEM, TADV and DDEP decrease to 5.35, 9.53, -5.52 and
-10.85 ppb/h. These reductions should be related with the process that the subtropical high moves
eastward and northward forced by Typhoon Utor. By quantifying the relative importance of each
process to $O_3$ formation, the IPR analysis provides a fundamental explanation for the synthetically
influence of the high pressure and the typhoon system, which has been discussed in Sect. 3.2 and
4.1, and further illustrates the exact mechanism.

Fig. 8b presents the result of IPR analysis for Hangzhou. During 4-15 August, VDIF and

CHEM are the major source of surface $O_3$ with the average contribution of 5.36 ppb/h for VDIF
and 10.97 ppb/h for CHEM, while TADV and DDEP are two important sinks for $O_3$ with the
average contribution of -9.63 ppb/h for TADV and -5.14 ppb/h for DDEP. Synthetically impacted
by Western Pacific subtropical high and Typhoon Utor, the mean contributions during the $O_3$
episode (from August 7 to August 12) for VDIF, CHEM, TADV and DDEP increase to 7.21, 12.61,
-11.51 and -5.92 ppb/h, respectively. The highest VDIF contribution occurs on August 10-11,
which may be attributed to the effect of typhoon's peripheral circulation, implying Typhoon Utor
also plays an essential role in the formation of $O_3$ pollution in Hangzhou. After Typhoon Utor
approach close enough to Hangzhou, the mean values of VDIF, CHEM, TADV and DDEP finally





decrease to 4.84, 10.08, -8.92 and -4.78 ppb/h, respectively. In a word, Hangzhou is located close
to Shanghai, so the temporal variations of VDIF, CHEM, TADV and DDEP in Hangzhou are
similar to those in Shanghai.

However, the similar variation pattern of VDIF, CHEM, TADV and DDEP occurring in

Shanghai and Hangzhou does not appear in Nanjing. As shown in Fig. 8c, the mean contributions
of VDIF, CHEM, TADV and DDEP to surface $O_3$ in Nanjing are 11.31, 9.55 -1.34 and -17.57
ppb/h during the whole period, while the values during 7- 12 August are 10.32, 10.70, -0.99 and
-18.42 ppb/h. There are no apparent fluctuations or sudden increases of these contributors during
the period from August 4 to 15, implying Nanjing is generally under the control of the Western
Pacific subtropical high and can hardly be affected by the typhoon system. As a typical city in the
northwest inland area of the YRD region (NIR), Nanjing is located far away from the sea, which
means it may not be easily affected by the weather system from the ocean.

Additionally, at the altitude of 500 m and 1500 m above Shanghai, Nanjing, and Hangzhou

(not shown), CHEM is also the major contributor to $O_3$ formation, with the values a litter lower
than those at the surface, suggesting that there are strong photochemical reactions in the whole
boundary layer of these YRD cities. In contrast, VDIF has an opposite effect in the middle of the
boundary layer, with the negative contributions for $O_3$ of -3.26 ppb/h in Shanghai, -2.37 ppb/h in
Hangzhou, and -3.21 ppb/h in Nanjing, respectively (not shown). The loss of $O_3$ at higher
atmospheric level caused by VDIF further proves the essential role of the downward vertical
movement in this $O_3$ episode.



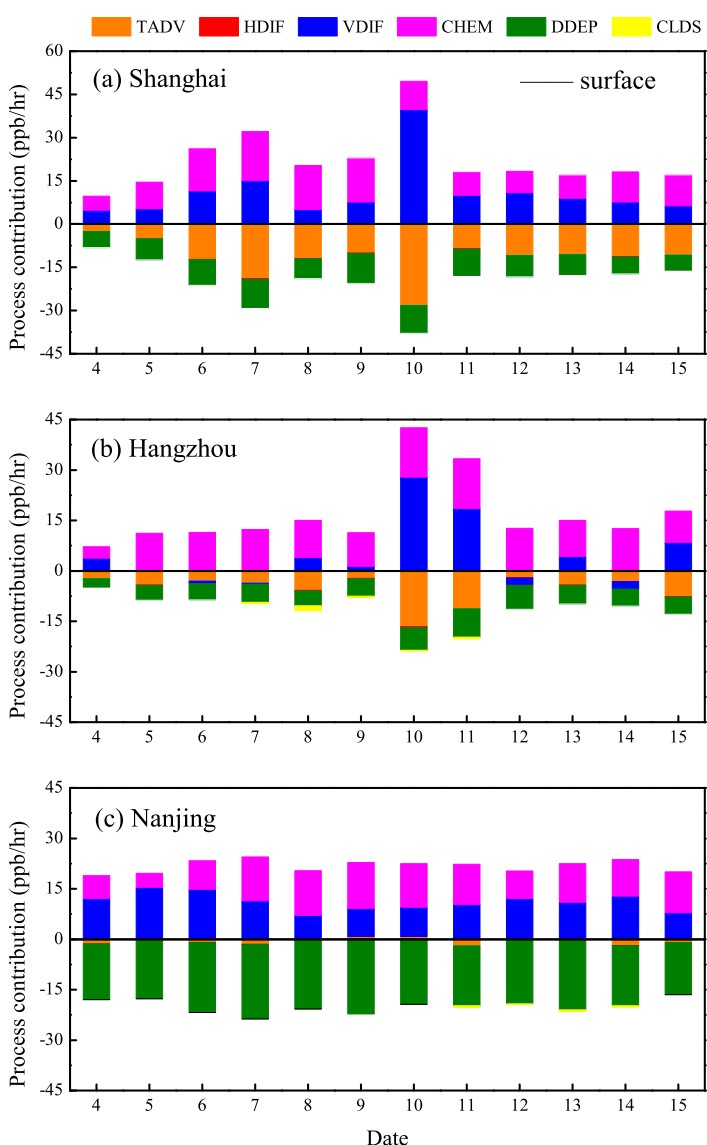

**Fig. 8. Variations of the daily mean values for the contributions of individual processes to O₃ formation in (a) Shanghai, (b) Hangzhou, and (c) Nanjing from 4 to 15 August 2013 at the surface layer. The contributors include the total advection (TADV), the horizontal diffusion (HDIF), the vertical diffusion (VDIF), the gas-phase chemistry (CHEM), the dry deposition (DDEP), and the cloud processes with the aqueous chemistry (CLDS).**

### 4.3.2 Spatial distribution of the contributors for the O₃ episode over the YRD region

Fig. 9 demonstrates the spatial distribution of the mean contributions of main processes



(TADV, VDIF, DDEP and CHEM) to the formation of this high-level $O_3$ episode at the lowest
modeling layer in domain 3. The modeling results from 7 to 12 August are averaged to provide the
mean values.
Similar to the results shown in Fig. 8, Fig. 9 illustrates that the vertical diffusion (VDIF) and
the gas-phase chemistry (CHEM) exhibit significant positive contributions to $O_3$ over the YRD
region and the surrounding areas during the high-level $O_3$ episode. The contributions of VDIF in
domain 3 (Fig. 9a) range from 5 to 25 ppb/h, with the high values (> 20 ppb/h) occurring in the
southeast coastal areas. For CHEM (Fig. 9b), the contributions vary within the range of 0-15 ppb/h,
with the high values over 10 ppb/h appearing in and around the big cities. As discussed above,
these regional positive contributions of VDIF and CHEM over domain 3 should be related to the
facts that the whole region is under the control of the Western Pacific subtropical high. With
respect to the higher contributions of CHEM in the urban areas, they should be attributed to the
spatial distribution of the emissions of $O_3$ precursors, which is also higher in the cities.
Furthermore, higher air temperature in the cities related with the urban heat island may enhance
the chemical reactions and form more $O_3$ in these areas as well.
As shown in Fig. 9c, DDEP is the main critical factor of the consumption of $O_3$, with the
negative contributions varying from 0 to -25 ppb/h over the modeling domain 3. Small values
usually occur on the water, which may be related with less air pollution over rivers, lakes and
oceans. High values can be found on land, especially in the southeast coastal areas. For the
contributions of TADV (Fig. 9d), the values in domain 3 range from -10 to 10 ppb/hr, with the
positive contributions generally occurring on land while the negative (consuming) ones appearing
on the water. The maximum positive contributions of TADV are usually found along the boundary
between the land and the water, which should be explained by the facts that the land-sea breeze
circulations can play an important role in the redistribution of the formed $O_3$.
In all, more active photochemical reactions and the vertical diffusion play a significant role in
the accumulation of surface $O_3$, and lead to the high-level $O_3$ pollution episode over the YRD
region. The major driving factor should be the Western Pacific subtropical high. Moreover, the
contributions of VDIF, DDEP and CHEM exhibit a similar spatial pattern with the high values
mostly concentrate in the southeast coastal areas, implying the Typhoon Utor also plays a
collaborative effect. The details and the processes of the synthetical effects of high pressure and




typhoon system have been discussed in Sect. 3.2, 4.2 and 4.3.1.

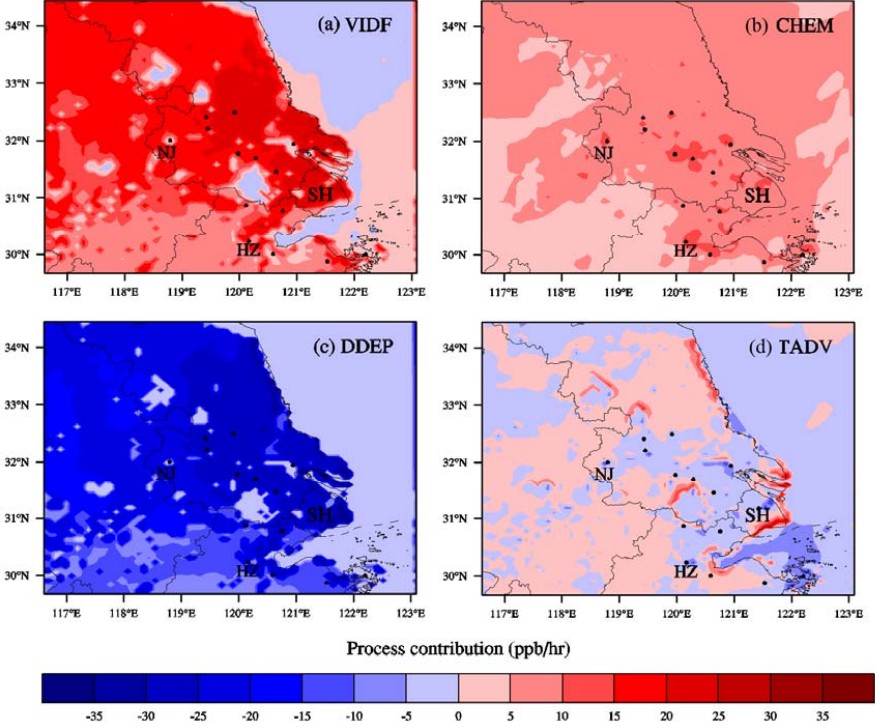

**Fig. 9. The contributions of main processes to O₃ formation over the YRD region, including (a) gas**
**chemistry (CHEM), (b) vertical diffusion (VDIF), (c) dry deposition (DDEP), and (d) total advection**
**(TADV). The values are averaged from 7 to 12 August 2013.**

**5. Conclusions**

By means of observational analysis and numerical simulation, the characteristics and the

essential impact factors of a typical regional continuous O₃ pollution over the YRD region is
investigated. Base on the observation data, it is found that this high-level O₃ episode lasts for
nearly a week from 7 to 12 August 2013, with the O₃ concentration exceeding the national air
quality standard in more than half of the cities over the YRD region. In the cities of Jiaxing,
Changzhou and Nantong, high O₃ concentrations can reach the values over 160 ppb. Fine weather
conditions, such as extremely high temperature, low relative humidity, and weak wind speed,
provide a favorable atmospheric environment for the complicated photochemical reactions and





help to form $O_3$. The analysis of weather systems and the modeling results from WRF/CMAQ all
illustrate that the continuous strong Western Pacific subtropical high is the leading factor of the
abnormal high temperature weather and the heavy $O_3$ pollution, by inducing more sinking air to
trap heat as well as air pollutants at the surface. The development of this episode is closely related
to the movement of Typhoon Utor as well. The temporal variations of the vertical wind velocity
and $O_3$ concentrations show that when the YRD region is at the front of moving typhoon system,
the downward airflow is enhanced in the boundary layer with fine weather, and thereby the air
pollutants are trapped and accumulated near the surface. Moreover, in the last stage of the $O_3$
episode, the activity of Typhoon Utor weakens the strength of the subtropical high and forces it to
retreat easterly and move northward, and the prevailing southeasterly surface wind related with the
approaching of Typhoon Utor contributes to the mitigation of the $O_3$ pollution.

The Integrated Process Rate (IPR) analysis implemented in CMAQ is specially carried out to

quantify the relative contributions of individual processes and give a fundamental explanation.
Over the YRD region, during the high-level $O_3$ episode from August 7-12, the vertical diffusion
(VDIF) and the gas-phase chemistry (CHEM) exhibit significant positive contributions to surface
$O_3$, with the high values over 20 ppb/h for VDIF and over 10 ppb/h for CHEM. The total
advection (TADV) can give the positive contribution on land and the negative contribution on the
water. The dry deposition (DDEP) is the major sink of surface $O_3$, while the contributions of
horizontal diffusion (HDIF) and cloud processes (CLDS) are quite small. To some extent, the
distribution pattern reflects the heterogeneity of emissions and the effects of weather system.
Influenced by the sinking air as well as the fine weather induced by the Western Pacific
subtropical high, the contributions of VDIF and CHEM to surface $O_3$ maintain the high values of
13.41 and 11.21 ppb/h for Shanghai, 7.21 and 12.61 ppb/h for Hangzhou, and 10.32 and 10.70
ppb/h for Nanjing, respectively. Moreover, on August 10-11, the cities close to the sea are
apparently affected by the periphery circulation of Typhoon Utor, with the contribution of VDIF
increase to 28.45 ppb/h in Shanghai and 19.76 ppb/h in Hangzhou. When the typhoon system
significantly weaken the high pressure system, the contributions of VDIF, CHEM, TADV and
DDEP decrease to a low level in all cities.

WRF-CMAQ model system shows a relatively good performance in simulation of the $O_3$

episode, with the simulated meteorological conditions and air pollutant concentrations basically in





agreement with the observations in most YRD cities. Our results in this study can provide an
insight for the formation mechanism of regional O₃ pollution in East Asia, and help to forecast the
O₃ pollution synthetically impacted by the Western Pacific subtropical high and the tropical
cyclone system.

**Acknowledgments**
This study was supported by the National Natural Science Foundation of China (41475122,
91544230, 41575145), the National Special Fund for Environmental Protection Research in the
Public Interest (201409008), EU 7th Framework Marie Curie Actions IRSES project REQUA
(PIRSES-GA-2013-612671), and the National Science Foundation of Jiangsu Provence
(BE2015151). The authors would like to thank Xiaoxun Xie for preliminary data processing, and
the anonymous reviewers for their constructive and precious comments on this manuscript.

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
