# Peer review of "Integrated studies of a regional ozone pollution synthetically"

_Atmospheric Chemistry and Physics, 2016_

## Referee Comment (RC1) · Anonymous Referee #2 · 29 Aug 2016

The paper studied a regional ozone pollution synthetically affected by subtropical high and typhoon system in the Yangtze River Delta region, China. The o3 pollution has been raised wide attention, and it's an interesting investigation and the manuscript is well written, the discussions are comprehensive, and the results are described clearly. I think that this paper deserves to be published in ACP.

---

## Referee Comment (RC2) · Anonymous Referee #3 · 8 Oct 2016

This study discussed the combined effects of two synoptic systems on O3 pollution in eastern China city clusters. Subtropical high was identified as the main cause of O3 episode, which was also influenced by a typhoon system. The case study was meaningful and to some extend representative of the typical O3 episodes in summer of eastern and southern China. However, many scientific and technical problems must be addressed before this paper is reconsidered to be accepted. Scientific problems: 1. Both VOCs and NOx are key precursors of O3, the authors only mentioned and analyzed the patterns of NO2 and CO. It was not enough to explain the spatial characteristics of O3 pollution. Even so, the authors did not well establish the possible rela-

tionships between O3 and NO2 and CO. Then, why to show them in Table 2. Suggest to add more O3 precursors and deepen the discussions. 2. Page 17, lines 410-413, the explanation to biases of O3 and NO2 were not convincing. The first sentence was meaningless. Although the second sentence described a common sense, it was not responsible for the discrepancies between simulation and observation. Note that the real conditions of O3 photochemistry are also non-linear. 3. Throughout the paper, the authors did not say why to select these four cities? Particularly for Wuxi, it was not a provincial capital or an extremely polluted city according to the information the authors provided. 4. Why were the modellings of wind speed and direction not validated with the observation data. They are closely related to the transport processes. 5. The northwest movement of high O3 centers from 10 to 12 Aug. was not obvious. On the basis of Fig.6, the center located at around 34N, 34N and 33N on these three consecutive days. Clarify and revise the discussions accordingly. 6. What was meant when vertical diffusion was used? How to explain diffusion process aggravated O3 pollution? Whether it was accurate to use vertical diffusion if the downward air flow transported high O3 to the surface. 7. Page 23, lines 541-549, evidences need to be provided to confirm whether NJ was influenced by typhoon. 8. In the section of process analyses, the observed and simulated weather conditions in terms of spatial and temporal patterns should be provided to aid the analyses. 9. Page 25, lines 586-591, why see-land breeze contributed positively to land O3 and negatively to oceanic O3? More supporting information are needed, such as the distribution map of O3 and wind fields. 10. Subtropical high is a common weather system dominating East Asia region in summer. Why did it only cause O3 episode in this period? Whether the effect was strengthened by the typhoon? On one hand, in the typhoon periphery, the strong downward flow stimulates O3 formation and suppresses air pollutants diffusion. On the other hand, had typhoon brought dirty air to the region from inland cities? Overall, the process analyses need to be more comprehensive. During the episode, when the subtropical high dominated and when typhoon dominated? Their combined effect was promotion or offset? Technical problems: 1. Too many grammatical errors in this paper, e.g. page

1, line 15 "is detected", line 16, "exceeding", "reaches", line 22 "abnormal strong", line 25 "worse air pollution". I cannot list all of them. Strongly suggest to correct the errors with the aid of a professional language correcting company. 2. Define the abbreviations at their first appearances, e.g. WRF-CMAQ. 3. Page 4, line94, what is the knowledge gap? 4. Past tense was suggested for the Introduction section, except for the common senses. 5. Website references needed to be added for the citations of meteorological and air pollutants data. 6. Page 9, lines 236-238, the function of water vapor and the evidence for its more abundance in Shanghai. 7. Page 17, line 395, WRF/CHEM or WRF-CMAQ? 8. Fig. 6, the legends need to be provided. Positive wind speeds with solid lines mean downward air flow? 9. Conclusion needs to be reorganized after the revision of the whole paper.

---

## Author Response (AR1)

**Response to the comments of Referee #1:**

The paper studied a regional ozone pollution synthetically affected by subtropical high and typhoon system in the Yangtze River Delta region, China. The o3 pollution has been raised wide attention, and it's an interesting investigation and the manuscript is well written, the discussions are comprehensive, and the results are described clearly. I think that this paper deserves to be published in ACP.

We would like to thank the referee for the valuable and affirmative comments of our manuscript.

**Response to the comments of Referee #2:**

This study discussed the combined effects of two synoptic systems on O3 pollution in eastern China city clusters. Subtropical high was identified as the main cause of O3 episode, which was also influenced by a typhoon system. The case study was meaningful and to some extend representative of the typical O3 episodes in summer of eastern and southern China.

However, many scientific and technical problems must be addressed before this paper is reconsidered to be accepted.

We appreciate the referee for the valuable and constructive reviews of our manuscript. We carefully revise the manuscript based on the following comments.

**Scientific problems:**

1. Both VOCs and NOx are key precursors of O3, the authors only mentioned and analyzed the patterns of NO2 and CO. It was not enough to explain the spatial characteristics of O3 pollution. Even so, the authors did not well establish the possible relationships between O3 and NO2 and CO. Then, why to show them in Table 2. Suggest to add more O3 precursors and deepen the discussions.

Thanks for the constructive comment. We agree that $NO_x$ and VOC are the most important precursors of $O_3$, and the possible relationships between $O_3$, VOC and $NO_x$ should be well established. So following the suggestion, the observed VOC records at an urban site in Shanghai (SAES, 31.17°N, 121.43°E) is added, and Section 3.1 is rewritten.

In the new revised manuscript, the time series of VOCs is shown in the new Fig. 3. The brief description of the site and the measurement method for VOC is added on lines 125-131. Meanwhile, in order to better discuss the $O_3$-VOCs-$NO_x$ relationship and reflect the basic characteristics of this $O_3$ pollution episode, the temporal variations of $NO_2$ in Shanghai, Hangzhou and Nanjing are also provided in the new Fig. 3. Based on the new Fig. 3, the temporal variations of $O_3$ and its precursors ($NO_2$ and VOC), as well as their internal links, are discussed. Please see lines 293-306 in the revised manuscript.

In the rewritten Section 3.1, we focus on the possible relationships of $O_3$, VOC and $NO_x$. So, the statistic data for CO in Table 2 and the simple discussion in the original manuscript are deleted. We have tried to find more VOC data to deepen our discussion. Unfortunately, only the records in Shanghai are available in this study, because the data of VOC in the Yangtze River Delta region are very limited and hard to get.

2. Page 17, lines 410-413, the explanation to biases of O3 and NO2 were not convincing. The first sentence was meaningless. Although the second sentence described a common sense, it was not responsible for the discrepancies between simulation and observation. Note that the real conditions of O3 photochemistry are also non-linear.

Thanks for the constructive comment. We carefully analyze the biases of $O_3$ and $NO_2$, and rewrite the words on lines 410-413 in the original manuscript.

In the new revised manuscript, we present three causes resulting in the biases, including the uncertainties in emissions of ozone precursors ("higher estimation of $NO_x$ emission in Shanghai leads to higher $NO_2$ and lower $O_3$ predictions, while lower $NO_x$ estimations in Nanjing and Hangzhou result in lower $NO_2$ and higher $O_3$ modeling results"), the overestimations in $WS_{10}$ and the negative biases in $T_2$, and some imperfections in the nonlinear chemical reactions coupled in CMAQ (especially the nocturnal chemistry). Please see lines 451-461 of the new revised manuscript.

3. Throughout the paper, the authors did not say why to select these four cities? Particularly for Wuxi, it was not a provincial capital or an extremely polluted city according to the information the authors provided.

Thanks for the constructive comment.

In the original manuscript, the meteorological and air quality observation data at the observation sites in Shanghai (SH, 31.40°N, 121.46°E), Hangzhou (HZ, 30.23°N, 120.16°E), and Nanjing (NJ, 32.00°N, 118.80°E) are used to validate the reliability of simulation by WRF/CMAQ. Unfortunately, we cannot get the meteorological records at the sites in Wuxi (31.62°N, 120.27°E). So, only the observed air quality data are adopted to evaluate the performance of CMAQ. Additionally, to reveal the roles of the individual physical and chemical processes involved in $O_3$ formation, we present the results of the IPR analysis for the typical cities such as Shanghai, Nanjing and Hangzhou. These cities (Shanghai, Nanjing, Hangzhou, and Wuxi) are all highly urbanized and industrialized, and suffer from severe $O_3$ pollution. For Shanghai, Nanjing and Hangzhou, they are provincial capitals and typical mega cities in YRD, and they also represent the cities in Southeast Coastal Region (SCR), Northwestern Inland Region (NIR), and Central Inland Region (CIR). For Wuxi, it is located between Shanghai and Nanjing, and also close to the Taihu Lake. So, we chose these cities for model validation and further analysis.

We are sorry that we did not emphasis why to select these cities in the original manuscript. In the new revised manuscript, we provide more information in Methodology to indicate our consideration. We rewrite the section "2.1 Observed meteorological and chemical data", and revise some words in the section "2.3 Integrated Process Rate (IPR) analysis method" and the section "2.4 Evaluation method". The comparison between the observed and the simulated $O_3$ concentrations in Wuxi is just a supplement for the evaluation of CMAQ. On account that the comparisons in Shanghai, Nanjing and Hangzhou are enough to prove the good performance of our simulations, the comparison for $O_3$ in Wuxi is deleted. Please see new figure 6 and lines 440-467 in the revised manuscript.

4. Why were the modellings of wind speed and direction not validated with the observation data? They are closely related to the transport processes.

Thanks for the constructive comment. We agree that the wind components "are closely related to the transport processes", and the modeling results of wind should be "validated with the observation data".

In the new revised manuscript, the validation for the modeling results of wind speed and direction is added in Section 4.1. Please see lines 417-425 and Table 3 of the revised manuscript.

5. The northwest movement of high O3 centers from 10 to 12 Aug. was not obvious. On the basis of Fig. 6, the center located at around 34N, 34N and 33N on these three consecutive days. Clarify and revise the discussions accordingly.

Thanks for the constructive comment. According to the suggestion, in order to clarify the northwest movement of high $O_3$ centers from 10 to 12 Aug., we replace the old Fig. 6 in the original manuscript by the new Fig. 7. The new Fig. 7 in the revised manuscript presents the daytime vertical wind velocity and the vertical distribution of $O_3$ from 116.5°E to 122.9°E along the latitude of 31.40°N (where Shanghai is located). Fig. 7d-f clearly show that the high value center of $O_3$ concentrations (above 90 ppb) moves westwards during August 10-12, implying that the peripheral circulation of Typhoon Utor can drive the air from the coastal areas to the inland areas. Accordingly, we revise the discussions. Please see lines 476-507 of the new revised manuscript.

6. What was meant when vertical diffusion was used? How to explain diffusion process aggravated O3 pollution? Whether it was accurate to use vertical diffusion if the downward air flow transported high O3 to the surface.

It is clear that due to the downward airflows (the upper parts of Fig. 7 in the new revised manuscript) dominated by the subtropical high and the typhoon system, the YRD region is under the stable, fine and hot weather condition, and the high concentrations (> 60 ppb) of ozone usually appear from the surface to 1.5 km height. However, as shown in Fig. 7 and 8 (revised in the new manuscript), the near-surface vertical velocity is much lower than that of higher altitudes. Especially in the planetary boundary layer near cities (< 1 km), lots of zero-velocity lines appear near the ground (Fig.8). This phenomenon may be related with the upward airflow caused by Urban Heat Islands. Thus, the maximum centers of $O_3$ occur near the surface below 500 m, and the vertical diffusion process plays a more important role in the accumulation of surface $O_3$ than the advection process. The role of vertical diffusion process in the $O_3$ episode is similar to that reported by Zhu et al. (2015).

Reference: Zhu, B., Kang, H. Q., Zhu, T., Su, J. F., Hou, X. W., and Gao, J. H.: Impact of Shanghai urban land surface forcing on downstream city ozone chemistry, J Geophys Res-Atmos, 120, 4340-4351, 10.1002/2014JD022859, 2015.

The above explanations are also added in the new revised manuscript to avoid the confusion from readers. Please see lines 476-558.

7. Page 23, lines 541-549, evidences need to be provided to confirm whether NJ was influenced by typhoon.

Thanks for the constructive comment.

Shanghai and Hangzhou are the cities more close to the sea, so they are easily affected by the typhoon system. Before August 13, $O_3$ concentrations in these cities exceed the national $O_3$ standard because of the synthetic effects of subtropical high and typhoon system. After August 13, the YRD region is totally under the control of the typhoon system, and thereby the hot weather is relieved and the $O_3$ pollution is mitigated. But for Nanjing, because it is far away from the coastal areas, it is hardly affected by the downward flow in the typhoon periphery. As shown in Fig.2 and Fig.8, the concentration of $O_3$ in Nanjing does not exceed the national $O_3$ standard. The increase of $O_3$ in Nanjing on August 12 (Fig.8c) should mainly be caused by the local photochemical reactions because the vertical movement below 2 km above Nanjing is dominated by upward airflows. In Fig. 9c, the IPR analysis also shows that Nanjing is not easily affected by the peripheral circulation of Typhoon Utor. After August 14 (the typhoon system significantly weakens the high pressure system), however, the decrease of $O_3$ concentration in Nanjing is influenced by the surface wind brought by the typhoon system.

According to this suggestion, we add more evidences in Fig. 8, including the temporal variations of the vertical wind velocity and the vertical distribution of $O_3$ above Hangzhou and Nanjing during August 7 to 12 2013. We also add some words (as shown above) to deepen our discussion. Please see lines 538-543 and 611-620 of the new revised manuscript.

8. In the section of process analyses, the observed and simulated weather conditions in terms of spatial and temporal patterns should be provided to aid the analyses.

Thanks for the constructive comment. We have added the relevant analysis in Section 4.3. Please see lines 562-610 of the new revised manuscript.

9. Page 25, lines 586-591, why see-land breeze contributed positively to land O3 and negatively to oceanic O3? More supporting information are needed, such as the distribution map of O3 and wind fields.

Thanks for the constructive comment. On account of the high-pressure system and so-caused sinking airflows in the YRD region, the background wind is relatively weak in comparison to the local atmospheric circulation, thus the sea breeze can easily bring more generated $O_3$ to the shore. We add some discussion in Section 4.3.2. Please see lines 662-665 of the new revised manuscript.

10. Subtropical high is a common weather system dominating East Asia region in summer. Why did it only cause O3 episode in this period? Whether the effect was strengthened by the typhoon? On one hand, in the typhoon periphery, the strong downward flow stimulates O3 formation and suppresses air pollutants diffusion. On the other hand, had typhoon brought dirty air to the region from inland cities? Overall, the process analyses need to be more comprehensive. During the episode, when the subtropical high dominated and when typhoon dominated? Their combined effect was promotion or offset?

Thanks for the constructive comment.

Although Western Pacific subtropical high (WPSH) is a normal influential system for the East China region in summer, but the WPSH is stronger and extends much farther west than normal in this period. The anomaly of the WPSH is the major and direct impacting factor for the hot weather (Peng et al., 2014). Besides, this $O_3$ episode is also influenced by the typhoon system. In the typhoon periphery, the strong downward flow may suppress air pollutants diffusion and cause contamination accumulation. It is not obvious that the typhoon brought dirty air to the region from inland cities. Instead, the clean marine inlet air brought by typhoon is advantageous to the dilution and diffusion of pollutants after the $O_3$ episode. We add the above explanation in the new revised manuscript.

To make the process analyses more comprehensive and answer the questions of reviewers, we rewrite the section 4.3.2 and new Fig. 11 in the new revised manuscript. Fig. 11 shows the differences of the contributions of main processes between the period of August 7-9 and August 10-12, which can quantitatively evaluate the role of the typhoon system in this severe high $O_3$ episode. It is clear that the abnormally strong WPSH dominated during the whole $O_3$ episode. Meantime, from August 10 to 12, the YRD is influenced by the typhoon system as well, because the changes in the contributions of VDIF, CHEM, DDEP, and TADV exhibit a similar spatial pattern with the high values mostly concentrating in the southeast coastal areas. Their combined effect is promoted during August 10 to 12.

Reference: Peng, J. B.: An Investigation of the Formation of the Heat Wave in Southern China in Summer 2013 and the Relevant Abnormal Subtropical High Activities, Atmospheric & Oceanic Science Letters, 7, 286-290, 2014.

**Technical problems:**
1. Too many grammatical errors in this paper, e.g. 1, line 15 "is detected", line 16, exceeding", "reaches", line 22 "abnormal strong", line 25 "worse air pollution". I cannot list all of them. Strongly suggest to correct the errors with the aid of a professional language correcting company.

Sorry for these grammatical errors in the original manuscript. The errors listed above are corrected as follows.

The words "is detected" on line 14 of the original manuscript are revised to "was detected". Please see line 13 in the new revised manuscript.

The word "exceeding" on line 16 of the original manuscript is revised to "exceeded". Please see line 15 in the new revised manuscript.

The words "abnormal strong" on line 22 of the original manuscript are revised to "abnormally strong". Please see line 21 in the new revised manuscript.

The words "worse air pollution" on line 25 of the original manuscript are revised to "worse air quality". Please see line 24 of the new revised manuscript.

Additionally, the authors asked an English native speaker (named Josh Powell from Georgetown University) to help improving the English.

2. Define the abbreviations at their first appearances, e.g. WRF-CMAQ.
Thanks for the constructive comment.

As suggested above, "CMAQ" on line 27 (where "CMAQ" occurs for the first time) of the original manuscript is replaced by "the Community Multi-scale Air Quality (CMAQ) Model". Please see lines 26-27 in the new revised manuscript.

The words "With the aid of the WRF/CMAQ" on lines 96 (where "WRF/CMAQ" appears for the first time) in the original manuscript are rewritten as "The WRF/CMAQ model system, which consists of the Weather Research and Forecasting (WRF) model and the Community Multi-scale Air Quality (CMAQ) Model, were used to reveal the exact formation mechanism." in the new revised manuscript. Please see lines 98-101 in the new revised manuscript.

"(LST)" on line 146 (where "LST" appears for the first time) in the original manuscript is changed to "(local standard time, LST)". Please see line 164 in the new revised manuscript.

"PBL" on line 154 (where "PBL" appears for the first time) in the original manuscript is revised to "planetary boundary layer". Please see lines 172-173 in the new revised manuscript.

"USGS" on line 157 (where "USGS" appears for the first time) in the original manuscript is replaced by "United States Geological Survey (USGS)". Please see line 176 in the new revised manuscript.

The sentence "The initial meteorological fields and boundary conditions are from NCEP FNL global reanalysis data with $1° \times 1°$ resolution" on lines 160-161 (where "NCEP" appears for the first time) in the original manuscript is rewritten as "The initial meteorological fields and boundary conditions are from 1° resolution global reanalysis data provided by National Center for Environmental Prediction (NCEP) ". Please see lines 178-180 in the new revised manuscript.

"(UTC)" on line 227 (where "UTC" appears for the first time) in the original manuscript is changed to "(Universal Time Coordinated, UTC)". Please see line 238 in the new revised manuscript.

"WRF-CMAQ" on line 642 of the original manuscript is revised to "WRF/CMAQ". Please see line 735 of the new revised manuscript. The definition has been given at its first appearances (on lines 98-101 in the new revised manuscript).

3. Page 4, line 94, what is the knowledge gap?

"Knowledge gap" means the principle that we do not know and need to be studied. The phrase "to fill the knowledge gap" was also used in some other papers (Ding et al.,2013; Xie et al., 2016). Here, the authors wanted to express that it is worth to investigate how the subtropical high and the typhoon system affect the formation of this regional $O_3$ pollution, and the results will help us to understand the important factors impacting O3 formation from the regional scale.

To avoid unnecessary misunderstanding, the words "To fill the knowledge gap and better understand the important factors impacting O3 formation from the regional scale, we perform an observational analysis to identify the temporal and spatial characteristics of the episode. With the aid of the WRF/CMAQ as well as the Integrated Process Rate analysis (IPR) coupled within CMAQ, numerical simulations are conducted to provide qualitative and quantitative analysis on the contributions of individual atmospheric processes." on lines 94-98 of the original manuscript are revised to "To better understand the important factors impacting O3 formation from the regional scale, we investigated the exact roles of these two typical weather systems in this pollution episode by using observational analysis and numerical simulations. The observational analysis was performed to identify the temporal and spatial characteristics of the episode. The WRF/CMAQ model system, which consists of the Weather Research and Forecasting (WRF) model and the Community Multi-scale Air Quality (CMAQ) Model, were used to reveal the exact formation mechanism. With the aid the Integrated Process Rate (IPR) analysis coupled in CMAQ, the qualitative and the quantitative analysis on the contributions of individual atmospheric processes were conducted as well.". Please see line 95-103 of the new revised manuscript.

4. Past tense was suggested for the Introduction section, except for the common senses.
Thanks for the constructive comment. As suggested above, several sentences in the Introduction section are revised to use the past tense.

The words "there is" on line 92 of the original manuscript are revised to "there was". Please see line 93 in the new revised manuscript.

The words "may be" on line 93 of the original manuscript are revised to "might be". Please see line 94 in the new revised manuscript.

The words "we perform" on line 95 and "are conducted" on line 97 of the original manuscript are deleted. "…, we investigated …", "The observational analysis was performed to …", "…, were used …" and "… were conducted" are added in the new revised manuscript. Please see lines 97-103 of the revised manuscript.

5. Website references needed to be added for the citations of meteorological and air pollutants data.
Thanks for the constructive comment. The website references for the citations of the meteorological and air pollutants data are added in the new revised manuscript. For the meteorological data, "The weather charts for East Asia are accessible from Korea Meteorological Administration (http://www.kma.go.kr/chn/weather/images/analysischart.jsp)", and "The hourly meteorological data at the observation sites of SH (31.40°N, 121.46°E) located in Shanghai, HZ (30.23°N, 120.16°E) in Hangzhou, and NJ (32.00°N, 118.80°E) in Nanjing can be obtained from the University of Wyoming (http://weather.uwyo.edu/wyoming/)". For the air pollutant data, "The in-situ monitoring data for the hourly concentrations of O3, CO, NO2, SO2, PM2.5 and PM10 can be acquired from National Environmental Monitoring Center (NEMC) (http://106.37.208.233:20035)". Please see lines 116-118 and 132-138 in the new revised manuscript.

6. Page 9, lines 236-238, the function of water vapor and the evidence for its more abundance in Shanghai.

Thanks for this constructive comment. In the new revised manuscript, we pay more attention to the chemical relations between $O_3$ and its precursors. Section 3.1 is rewritten. Water vapor is not discussed, and the words on lines 236-238 of the original manuscript are changed to "It seems that O3 concentrations are higher in the cities around Shanghai, where the concentrations of O3 precursors (shown in Table 2) are more adequate as well.". Please see lines 285-287 of the new revised manuscript.

7. Page 17, line 395, WRF/CHEM or WRF-CMAQ?

Sorry for this clerical error. The word "WRF/CHEM" on line 395 of the original manuscript is revised to "WRF/CMAQ". Please see line 436 of the new revised manuscript.

8. Fig. 6, the legends need to be provided. Positive wind speeds with solid lines mean downward air flow?

Thanks for the constructive comment. The dotted lines show the negative wind speeds and represent downward airflow, while the solid lines show the positive vertical wind speeds and zero vertical velocity. In the new revised manuscript, the legends and the necessary explanation are added to Fig. 7 and 8. Please see lines 512-514 and 548-550 of the new revised manuscript.

9. Conclusion needs to be reorganized after the revision of the whole paper.

As suggested above, the conclusion is reorganized and revised after the revision of the whole paper. Please see lines 705-740 of the new revised manuscript.

[revised manuscript text omitted]

variable. The simulated wind speeds are a little higher than observed results, but still reasonable.

Owing to the vectorial nature of wind, statistical comparisons for wind direction are shown excluding mean values of observation and simulation data, with the values of NMB at SH, NJ and

HZ site being 3.53%, -1.57% and -0.13%, respectively. However, Aaccording to the relevant studies (Li et al., 2012; Liao et al., 2015; Xie et al., 2016a), this level of over- or under-estimation is still acceptable. The wind components are closely related to the transport processes. As shown in Table 3, our modeling results of wind speed and direction basically reflect the characteristics of wind fields. For 10-m wind speed ($W_{spd10}$), R is 077 at SH, 0.74 at NJ, and 0.75 at HZ, respectively. Though the values of NMB (1.53%, 5.92%, and 9.21%) and RMSE (2.18, 2.41 and

2.39) display that the simulated wind speeds are a little overestimated, the biases are still reasonable and acceptable. For 10-m wind direction ($W_{dir10}$), the simulated values also fit the observation records well, with the R values of 0.63 at SH, 0.57 at NJ and 0.58 at HZ. Comparing the mean values from SIM and OBS, we can find that WRF model generally simulates the prevailing wind direction during this period. In summary, the abovementioned performance statistics numbers basically illustrate that the WRF simulation can reflect the major characteristics of meteorological conditions during of this O$_3$ episode, and the meteorological outputs can be used in the pollutant concentration simulation.

**Table 3. Comparisons between the simulations and the observations at Shanghai, Nanjing and Hangzhou**

**stations during August 4-15 2013.**

[revised manuscript text omitted]

~~the vertical $O_3$ concentration distribution and vertical velocity at Nanjing site show differences. The $O_3$ concentrations below boundary layer is substantially lower than that of Shanghai and Hangzhou. Besides, from August 7 to 10, vertical velocity is found extremely weak near the ground, as massive zero velocity lines appear below 500 m. From August 11 to 12, although the $O_3$ concentration is increasing as the high center is upper at the altitude of around 1 km. But, the vertical movement below 2 km is dominated by upward airflow, thus the surface $O_3$ pollution is relieved with the $O_3$ concentration lower than the national standard (Fig. 2).~~

(a) Shanghai (b) Hangzhou (c) Nanjing

Ozone Concentration (ppb)

[revised manuscript text omitted]